# Calumenin, a Ca^2+^ Binding Protein, Is Required for Dauer Formation in *Caenorhabditis elegans*

**DOI:** 10.3390/biology12030464

**Published:** 2023-03-17

**Authors:** Kyung Eun Lee, Jeong Hoon Cho, Hyun-Ok Song

**Affiliations:** 1Department of Infection Biology, Wonkwang University School of Medicine, Iksan 54538, Republic of Korea; 2Department of Biology Education, College of Education, Chosun University, Gwangju 61452, Republic of Korea; 3Department of Biomedical Science, Graduate School, Wonkwang University, Iksan 54538, Republic of Korea; 4Institute of Wonkwang Medical Science, Wonkwang University, Iksan 54538, Republic of Korea

**Keywords:** cuticle, dauer, calcium-binding protein, calumenin, *C. elegans*

## Abstract

**Simple Summary:**

*C. elegans* dauer is a form of phenotypic and developmental plasticity that induces reversible developmental arrest upon adverse environmental cues. This is well conserved among nematodes, including in human-infective species that cause filariasis, which is a major neglected tropical disease with considerable morbidity throughout tropical countries. The infective larval stage of parasitic nematodes is equivalent to the dauer stage of free-living nematodes; thus, it is important to elucidate the unknown molecular mechanisms behind dauer development. In this sense, to understand the infective mechanism of human-infective nematodes, it is worthwhile to study the novel function of calumenin in *C. elegans* dauer formation. In this study, we first found that calumenin is strongly expressed in body-wall muscle and AIA interneurons at the dauer stage. In addition, the functional loss of calumenin led to failures in dauer formation induced by either single pheromone compounds or crude pheromone extracts, which are generally used to induce dauer formation in *C. elegans*. Based on the transcriptional and translational expression of calumenin at the dauer stage, our results suggest that calumenin may function in signaling pathways that regulate dauer formation by a yet unknown mechanism.

**Abstract:**

*Caenorhabditis elegans* can adapt and survive in dynamically changing environments by the smart and delicate switching of molecular plasticity. *C. elegans* dauer diapause is a form of phenotypic and developmental plasticity that induces reversible developmental arrest upon environmental cues. An ER (endoplasmic reticulum)-resident Ca^2+^ binding protein, calumenin has been reported to function in a variety of malignant diseases in vertebrates and in the process of muscle contraction–relaxation. In *C. elegans*, CALU-1 is known to function in Ca^2+^-regulated behaviors (pharyngeal pumping and defecation) and cuticle formation. The cuticles of dauer larvae are morphologically distinct from those of larvae that develop in favorable conditions. The structure of the dauer cuticle is thicker and more highly reinforced than that of other larval stages to protect dauer larvae from various environmental insults. Since the *calu-1*(*tm1783*) mutant exhibited abnormal cuticle structures such as highly deformed annuli and alae, we investigated whether CALU-1 is involved in dauer formation or not. Ascaroside pheromone (ascr#2) and crude daumone were used under starvation conditions to analyze the rate of dauer formation in the *calu-1(tm1783)* mutant. Surprisingly, the dauer ratio of the *calu-1*(*tm1783*) mutant was extremely low compared to that of the wild type. In fact, the *calu-1*(*tm1783*) mutants were mostly unable to enter diapause. We also found that *calu-1* is expressed in body-wall muscle and AIA interneurons at the dauer stage. Taken together, our results suggest that CALU-1 is required for normal entry into diapause in *C. elegans*.

## 1. Introduction

Animals, including *Caenorhabditis elegans*, encounter highly dynamic and complex challenges such as limited food supplies, ambient temperature changes, and overcrowded populations in the natural environment. Under these conditions, *C. elegans* can adapt and survive by the smart and delicate switching of developmental plasticity [1]. *C. elegans* dauer is a form of phenotypic and developmental plasticity that induces reversible developmental arrest upon environmental cues [1,2]. *C. elegans* L1 (1st larval stage) larvae usually select one of two developmental pathways in response to environmental stimuli. In favorable conditions with plentiful food and a low population density, L1 larvae select the reproductive pathway and continue to develop through the other three sequential larval stages (L2, L3, and L4), and finally reach the adult stage. However, in harsh conditions with scarce food and a high population density, L1 larvae are triggered to enter diapause and arrest development as dauer larvae, which is an alternative L3 stage [1,3]. These specialized larvae do not feed, but are motile and explore new environments that may be favorable to them. Dauer larvae also have thicker and more highly reinforced cuticles than other larval stages, protecting them from various environmental insults [1,4,5,6]. Dauer formation is usually triggered by sensory cues that are mainly pheromones [7,8]. Seven amphid chemosensory neurons (ASI, ADF, ASG, ASJ, ASK, AWA, and AWC) have been known to mediate diapause entry [9,10,11,12] via multiple distinct pathways such as the guanylyl cyclase pathway, TGF-β pathway, and insulin-like pathway [13,14,15,16,17,18].

Calumenin is an ER-resident Ca^2+^ binding protein belonging to the CREC family [19] and has been reported to function in a variety of malignant diseases [20,21,22,23] and in the process of muscle contraction–relaxation in vertebrates [24,25]. In *C. elegans*, calumenin is known to function in Ca^2+^-regulated behaviors such as pharyngeal pumping and defecation [26]. It is also known to be involved in the formation of normal cuticle structures [26]. The loss of calumenin (*calu-1*) function in nematodes leads to cuticle defects, resulting in a semi-dumpy and semi-roller morphology in worms [26]. In addition, the *calu-1* mutant (*calu-1*(*tm1783*)) exhibits abnormal cuticle structures, showing irregular annuli and deformed alae [26]. Appropriate cuticle formation is very important during development, as cuticle morphology undergoes special changes during the dauer stage. Thus, we tested whether *calu-1* does or does not contribute to appropriate dauer formation. The results showed that *calu-1* is expressed in both the neurons and effector tissues that mediate dauer entry. In addition, the *calu-1*(*tm1783*) mutant is dauer-defective in response to both single purified dauer pheromones and crude pheromone extracts. These results suggest that CALU-1 is required for normal entry into diapause in *C. elegans*.

## 2. Materials and Methods

### 2.1. C. elegans Strains and Cultivation

The following strains were used in this study: Bristol N2 (wild type), *calu-1*(*tm1783*) X, CF1038 *daf-16*(*mu86*) I, CB1370 *daf-2*(*e1370*) III, BC12836 *dpy-5*(*e907*)*/dpy-5*(*e907*), and sIs11268 (rCes M03F4.7:GFP + pCeh361). *calu-1*(*tm1783*) X was obtained from the National BioResource Project, Japan and others were obtained from the Caenorhabditis Genetics Center (CGC). Worms were grown on Nematode Growth Media (NGM) seeded with *Escherichia coli* OP50 as a food source, according to standard methods [27].

### 2.2. Temporal Expression of calu-1 by Quantitative RT-PCR

RNA was extracted from six developmental stages: Eggs of N2 were collected at 20 °C for 2 h and further incubated at 20 °C for 14, 29, 38, 62, and 67 h to obtain L1s, L2s, L3s, L4s, and young adults, respectively [28]. For L2d predauers, the *daf-2*(*e1370*) mutant was used because it shows a constitutive dauer phenotype at a permissive temperature of 25 °C [29]. Eggs of the *daf-2*(*e1370*) mutant were collected at 20 °C for 2 h and incubated at 25 °C for 12 h [29]. Total RNA was then prepared from each synchronized animals using a Total RNA isolation kit (Marchery-Nagel) and used for cDNA synthesis (Roche, Transcriptor first strand cDNA synthesis kit). The subsequent amplification of the *calu-1* gene was performed with the forward primer (5’-GGT GAA CAT TTC AAG GGA AAG GAA CAT GAC-3’) and the reverse primer (5’-ATG TGA TCC TTG AGC TCG TTC TCC TCG-3’). The actin gene was amplified as an internal control with the forward primer (5’-GAG GCC CAA TCC AAG AGA GGT ATC CTT AC-3’) and the reverse primer (5’-TTC ACG GTT AGC CTT TGG ATT GAG TGG-3’). The amplification was conducted using 100 ng cDNA templates and 500 nM primers, and SYBR green reagent (Agilent, Brilliant III Ultra-Fast SYBR Green QPCR Master Mix) in an Agilent AriaMx Real-time PCR machine with the following conditions: 95 °C, 3 min, 1 cycle; 95 °C, 5 s, 60 °C, 10 s, 40 cycles (amplification); 95 °C, 30 s, 65 °C, 30 s, 95 °C, 30 s, 1 cycle (melting curve analysis).

### 2.3. Construct and Microscopy

To construct pSG1 (*calu-1* promoter::full length cDNA of *calu-1*::GFP), an approximately 2 kb-long promoter was amplified from pAN298 (*calu-1* promoter::pPD95.79, kindly provided by Dr. Joohong Ahnn) with the forward primer (5’-GAA ATA AGC TTG CAT GCA ACA TAT GAG C-3’) and the reverse primer (5’-CAA TCC CGG GGT CCA GAT CCT ACT A-3’). Then, the full length of the *calu-1* cDNA was amplified from pAN276 (full length cDNA of *calu-1*::pGEX 4T-1, kindly provided by Dr. Joohong Ahnn) with the forward primer (5’-CCG GAA TTC CCG GGA TGA AGG TTC TT-3’) and the reverse primer (5’-GGC CGC TCG GGT ACC AGC TCG GCT-3’). Both amplified products were sequentially subcloned into a promoter-less GFP vector, pPD95.79 (kindly provided by Dr. Andrew Fire), using the Sph I, Xma I, and Kpn I sites. The resulting plasmid was microinjected into wild type N2 animals, and then, transgenic animals that stably isolated the transgene over several generations were selected. After stable transgenic lines were obtained, worms were immobilized on a 2% agarose pad by levamisole and visualized using a super-resolution confocal microscope (LSM980, Carl Zeiss, Oberkochen, Germany) at the Core Facility for Supporting Analysis and Imaging of Biomedical Materials in Wonkwang University, supported by the National Research Facilities and Equipment Center.

### 2.4. Dauer Formation Assay

The synthetic stock solution of ascaroside pheromone 2 (ascr#2, kindly provided by Dr. Jun Young Park and Dr. Young-Ki Paik at Yonsei Proteome Research Center) was prepared by adding 1 mL absolute ethanol, as described in [30,31]. The crude pheromone was extracted according to a previous report, with minor modifications [32]. Briefly, large numbers of worms were cultured for several days, supplemented with a bacterial diet to increase population density. This cultivation induces the worms to produce many dauer pheromones. The crude pheromone dose that yielded 50% dauer formation was determined by generating a pheromone dose response curve [33]. The assay was performed based on a previous study, with modifications [31]. Briefly, 38 μM ascr#2 or crude daumone-supplemented NGM (without peptone or tryptone) were seeded with 160 μg heat-killed *E. coli* OP50. Heat-killed *E. coli* was prepared by heating the bacterial suspension at 95 °C for 30 min, with vigorous agitation every 5 min—as described in [30]. Eggs were collected on the assay plates for 1 h at 20 °C and grown on the assay plates for 3 days at 25 °C. The resultant dauers were counted. Dauers’ morphological characters (thin body shape and buccal plug, if necessary) and SDS resistance were determined.

### 2.5. Statistical Analysis

Graphpad Prism (Version 5.00 for windows, GraphPad Software, San Diego, CA, USA) was used to produce graphs and to perform the statistical analysis. The *p*-values were calculated using the paired *t*-test with 95% confidence intervals.

## 3. Results

### 3.1. Calu-1 Is Expressed throughout Development—Including the Dauer Stage

RT-PCR experiments showed that *calu-1* mRNA is present in the dauer stage as well as at all reproductive larval and adult stages (Figure 1). The actin gene (*act-1*) was amplified as an internal control to check the quality of RNA and cDNA. The expression level of *calu-1* was relatively lower than that of *act-1,* but its expression was maintained throughout all reproductive stages of development and at the predauer larval stage, with only slight fluctuations.

### 3.2. Calu-1 Is Expressed in Both Neuron and Effector Tissues That Mediate Dauer Entry

*calu-1* has been shown to be expressed in the pharynx, intestine, muscle and hypodermis during reproductive development [26]. In this study, *calu-1* was also found to be strongly expressed in body-wall muscle and AIA interneurons at the dauer stage (Figure 2a–d). It is clearly visible that the body-wall muscles surround the body of the dauer animal (Figure 2a); one of four quadrants of the body-wall muscles were visible along the dauer body’s anterior–posterior axis (Figure 2b). *calu-1* was expressed in AIA interneurons at both the dauer and adult stages (Figure 2c–f). The cell body of the AIAL and AIAR were located close to each other at the ventral side of the terminal bulb of the pharynx (Figure 2c,d) and the dendrites formed the nerve ring at the dorsal midline (Figure 2c,d,f), as previously reported [34]. In addition, distinct ER expression patterns for *calu-1* were observed in the hypodermis (Figure 2g) and body-wall muscle of adults (Figure 2h).

### 3.3. The calu-1(tm1783) Mutant Is Dauer-defective

The *calu-1*(*tm1783*) mutant has been shown to exhibit multiple defects such as a slightly dumpy body, uncoordinated movement, reduced fertility, and slow growth compared to the wild type [26]. Since the *calu-1*(*tm1783*) mutant also shows severe cuticle defects [26] and as the *calu-1* gene is found to be expressed at the dauer stage (Figure 1 and Figure 2), we questioned whether *calu-1* is involved in the regulation of dauer commitment. To answer this question, the dauer formation ratio was addressed in the *calu-1*(*tm1783*) mutant using both ascaroside#2 (ascr#2) and crude pheromones (Figure 3a). Approximately 50% of N2 (wild type) animals successfully entered diapause under the experimental condition (Figure 3b,c). The dauer-defective mutant *daf-16*(*mu86*) failed to enter diapause, as expected (Figure 3b,c). Surprisingly, the *calu-1*(*tm1783*) mutant showed a similar phenotype to that of the *daf-16*(*mu86*) mutant (Figure 3b,c); it failed to form dauers with either a single purified dauer pheromone or the crude pheromone extract (Figure 3b,c).

## 4. Discussion

Calumenin is an ER-resident protein that contains five (for *C. elegans*) to six (for human) EF-hand motifs with a relatively low affinity to Ca^2+^ [26,35]. In vertebrates, calumenin has been implicated in various malignant diseases, although there is no simple direction of regulation [20,21,22,23]. Instead of the typical ER retention signal sequence, HDEL, calumenin contains a C-terminal HDEF sequence, which is not efficient for retention in the ER [35]. Thus, this causes calumenin to be distributed throughout the secretory pathway; indeed, it has been found to be secreted from cells [36,37,38]. In addition, calumenin has been found to be located in membranes and interact with various membrane proteins including the ryanodine receptor [39] and cardiac Ca^2+^-transporting ATPase [25], or extracellular matrix proteins such as fibulin [40]. *C. elegans* has one homolog of calumenin and has been reported to function in various Ca^2+^-regulated behaviors [26]. *C. elegans* calumenin has also been proposed as a potential drug target for human-infective nematodes (filarial worms) due to its implication in normal cuticle development [41].

As such, calumenin has been reported to play a role in various cellular functions. Here, we briefly report a novel function of calumenin in the formation of the dauer—a form of developmental adaptation to the environment in nematodes. First of all, we confirmed that both the calumenin gene (*calu-1*) and protein (CALU-1) are expressed in the dauer stage—especially in cellular locations related to dauer entry such as the interneuron, muscle, and hypodermis. As reported and expected, CALU-1 was found to be expressed in the muscle and hypodermis during reproductive development (Figure 1 and Figure 2e–h). Interestingly, in this study, we newly found that CALU-1 is expressed in the remodeled muscles as well as the AIA interneuron of dauers (Figure 2a–d). Despite the limitation that an AIA co-marker was not used in this study, a recent study of single-cell RNA sequencing in the *C. elegans* nervous system revealed that the *calu-1* transcript is expressed in AIA neurons, supporting our results [42]. The AIA is one of the first-layer amphid interneurons, receiving extensive synaptic inputs from sensory amphid neurons [43] and regulating various behavioral responses such as locomotion, chemotaxis, and learning [44,45,46,47,48,49]. A recent report revealed that AIA interneurons integrate external sensory cues into effector tissue signaling pathways via neuropeptidergic propagation to determine larval developmental fate [33]. In other words, AIA interneuron-derived FLP-2 neuropeptide signaling is found to promote reproductive growth and AIA activity is inhibited by pheromones to induce diapause entry [33]. Thus, this provides the possibility that *calu-1* may be involved in this process, as *calu-1* is expressed in AIA interneurons and large precursors of neuropeptides are processed by multiple proteolytic enzymes in the ER where CALU-1—a Ca^2+^ binding protein—usually functions [50]. In addition, it is also possible that *calu-1* is involved in the reconstruction of the intestine, cuticle, and muscle as it is also expressed in the ER of these effector tissues. In this sense, the insulin pathway and the TGF-β pathway should be considered, as they are the major signaling pathways regulating dauer formation in endocrine cells. Further studies are needed to elucidate the molecular mechanisms of how calumenin functions for normal entry into diapause in *C. elegans*.

## 5. Conclusions

Dauer formation is one strategy for nematodes to adapt to and survive highly dynamic and complex environmental insults. However, it can be maladaptive if an incorrect perception of environmental cues and incorrect integration of downstream effector tissue signaling occur as dauer larvae can fully re-enter the trajectory leading to reproductive development within 20 h. Thus, highly sophisticated control over these complex and distributed regulatory networks is critical. In this study, the role of calumenin was highlighted in dauer formation. Since the structure of the dauer cuticle is reorganized during dauer stages to protect them from various environmental insults, and as the *calu-1*(*tm1783*) mutant exhibits abnormal cuticle structures, it is highly plausible that calumenin may function in the reconstruction of this effector tissue. In addition, calumenin may function in neuropeptidergic signal propagation in AIA interneurons to determine larval developmental fate, as it is expressed in the AIA interneuron at the dauer stage. Although further studies are needed to elucidate the exact molecular mechanisms of how calumenin functions for normal entry into diapause in *C. elegans*, this study indicates that calumenin is a novel regulator in the pathways leading to dauer development in *C. elegans*.

## Figures and Tables

**Figure 1 biology-12-00464-f001:**
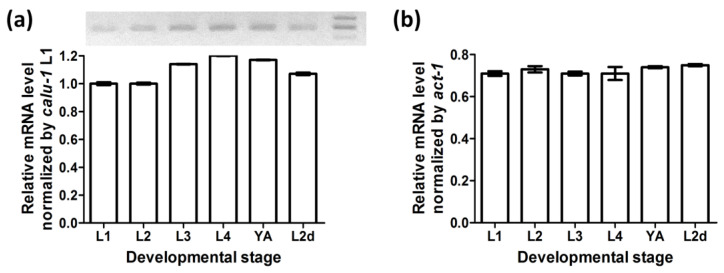
Expression of *calu-1* at developmental stages. Three independent assays were performed using cDNA libraries generated from wild type N2 (L1, L2, L3, L4, and YA) or *daf-2(e1370)* mutants (L2d). (**a**) Relative mRNA level of *calu-1* by L1 expression level to show differential expression levels of *calu-1* at developmental stages. Representative gel image of quantitative real time RT-PCR is shown above the graph. Each well corresponds to a PCR product for each developmental stage. Size markers on the right of the image indicate 250, 200, and 150 bp from top to bottom. (**b**) Relative mRNA level of *calu-1* normalized by internal control (*act-1*). L1, first larval stage; L2, second larval stage; L3, third larval stage; L4, forth larval stage; YA, young adult stage; L2d, L2 predauer stage.

**Figure 2 biology-12-00464-f002:**
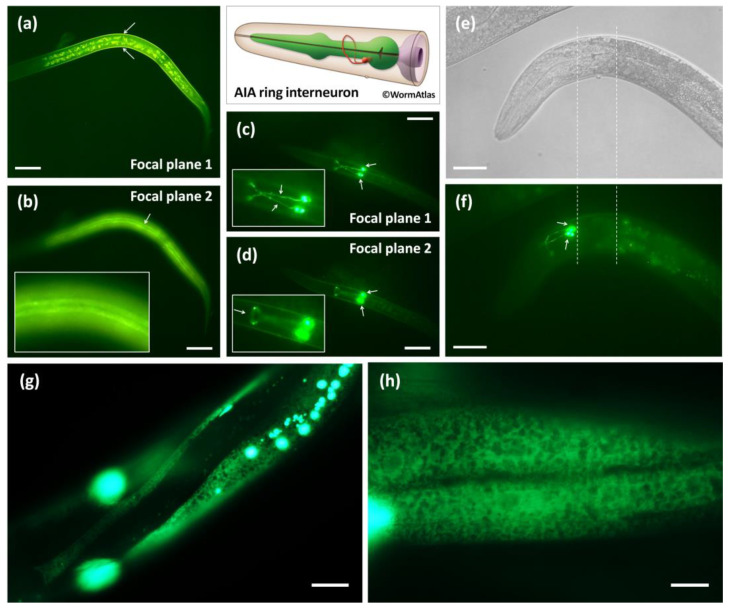
In vivo expression of *calu-1* in dauers and adults. (**a**,**b**), transgenic animals expressing the promoter of *calu-1* fused with GFP (transcriptional reporter); (**c**–**h**), transgenic animals expressing the full-length cDNA of *calu-1* fused with GFP, driven by its own promoter (translational reporter). (**a**) It is visible that body-wall muscles surround the body of dauer larvae (indicated by arrows). (**b**) One of four quadrants of the body-wall muscles is visible (indicated by an arrow and a magnified image) (**c**,**d**) Cell bodies (AIAL and AIAR, indicated by arrows) and dendrites of the AIA interneuron are clearly visible. The dendrites are more visible in ((**c**); indicated by arrows in a magnified image) and the nerve ring of the dendrites is only visible in ((**d**); indicated by an arrow in a magnified image). Normaski (**e**) and GFP (**f**) images of a young adult. The terminal bulb of the pharynx is indicated by two dashed lines (**e**,**f**). (**f**) AIA interneuron is clearly visible, with cell bodies (indicated by arrows) and dendrites. *calu-1* is found in the ER of the hypodermis (**g**) and the muscle (**h**) of adults. Scale bars, 20 μm, 10 μm only in (**h**).

**Figure 3 biology-12-00464-f003:**
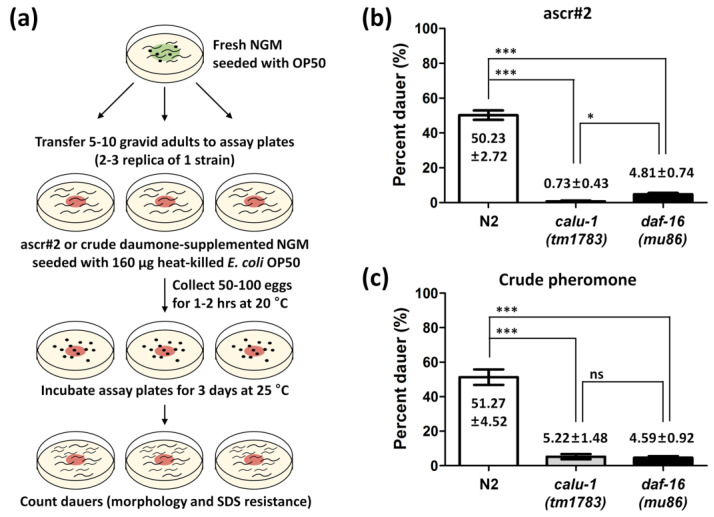
Dauer formation assay of the *calu-1*(*tm1783*) mutant. (**a**) Schematic diagram of the dauer formation assay. Four independent assays were performed, each in duplicate or triplicate, using ascr#2 (**b**) or crude dauer pheromone (**c**). Mean ratios with SEM (standard error of mean) are represented in bar graphs. ***, *p* < 0.001; *, *p* < 0.05; ns, not significant.

## Data Availability

Not applicable.

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
