# Peer review of "Calumenin, a Ca2+ Binding Protein, Is Required for Dauer Formation in Caenorhabditis elegans"

_biology, 2023, doi:10.3390/biology12030464_

Round 1

Reviewer 1 Report

The manuscript entitled “Calumenin, a Ca2+ binding protein, is required for dauer formation in Caenorhabditis elegans” presents relevant information about the role of Calumenin in the formation of the dauer diapause. However, some sections of the presented data can be improved. For this reason, I consider that this manuscript needs changes to be considered for its publication in this journal.

Major Revisions:

Line 138. Why the expression of calu-1 was lower than the housekeeping? it is advisable to use another housekeeping

Figure 1. Why the graph axis is in Ct values? it is usually express in Relative mRNA level.

The type of worms employed in this assay should be in the figure, are daf-2 mutants?

Line 184. Delve into the discussion of the own results, for example the results of RT-qPCR and the mutant strains.

Minor comments:

Line 27. Explain ER abbreviation.

Line 81. It is necessary to add the strain of calu-1(tm1783) X, I didn´t find this mutant in CGC, Where was this strain obtained from?

Line 82. It is necessary to add the strain of daf-16(mu86) I

Line 91. Explain why daf-2 mutants were selected for this essay

Line 94. cule-1 cursive

Line 120 and 121. Avoid two close “as described”. Give more details of the process of preparation of ascaroside pheromone 2 and Crude pheromone

Line 125. How were you sure E. coli was dead? explain how killed the bacteria

Line 133. Remove a full stop

Author Response

Please find the attache file.

Reviewer 2 Report

The dauer of C. elegans has been studied very well, however, the function of Calumenin on dauer formation has not been performed before. Here, the author quantified the gene calu-1 expression level among its different life stages and showed the potential expression site of the nematode body. Based on mutant of calu-1(tm1783), the author indicated that calu-1 is involved in dauer formation of C. elegans. However, there were some comments on this manuscript as below.

1. Calumenin is an ER-resident protein. FISH (fluorescence in situ hybridization) is more suitable for protein location. I am not sure why the author used promoter of Calu-1 with GFP and cDNA of this gene for the expression location of gene. Meanwhile, in the methods, why the cDNA of calu-1 was cloned into pPD95.79. The vector with calu-1 promoter and promoterless GFP were enough for functional location.

2. The methods were incomplete. For example, the methods of qRT-PCR, how to get transgenic lines.

Other comments:

Figure1, the Ct value was fluctuant with different cDNA concentration, primers. Different primers have different amplication efficiency. So I think it make no sense for comparing the Ct values between act-1 and calu-1. Actin gene is a reference gene for correct different cDNA concentration of different samples. So the Ct value would not be shown in the figure results and the calu-1 expression should be normalized by the act-1 value at the same life stage (figure 1A and B). I am confused of the result of figure 1C. The same reason as before, there were no sense for comparing the expression between act-1 and calu-1 of different life stages. Additionally, the intensity level of markers of the two gel were different, how to compare these two picture.

Table 1, it is a repeat data of figure 1, should be deleted

Figure 2, the “d” is missing. The scale bars of figure 2a, b, c, and d is 20um. But why the nematode body of figure 2d (the wrong c’) seem bigger than the others so much. I think it is no sense for your study to show the fluorescence figure with different plane. So the blurry figure should be deleted, or instead by visible images. “Transcriptional” at line 161 and “Translational” at 162, what is the difference between this two words.

Line 167, I want to know the development of tm1783. By absent of a gene calu-1, how were the life cycle, body development or movement of this mutant.

Figure 3, by your description as figure 3b and d, four independent assays were performed for the final analysis of figure 3c and d. So the intermediate data should be deleted (figure 3b and d). If you want to show this experiment, you can add some sentences in methods or use scatter diagram to show your result.

Reviewer 3 Report

Kyung Eun Lee et al. Reported calu-1 is required for dauer formation in C. elegans. This paper measured the mRNA expression of calu-1 in different stages and protein expression of calu-1 by GFP marker. Then the authors tested the dauer formation in Wild-type and calu-1 knockout worms. However, this paper missing lots of information and needs to compare their results with other publications before publishing. This paper also missing the mechanism of how calu-1 regulates dauer formation.

Major points:

1. Fig.1 mRNA expression. The authors seems do not know how to use the C. elegans website: wormbase. The FPKM value of act-1 are 1226 at L1, 1337 at L2, 2211 at L3,1054 at L4, 739 at young adult. The FPKM value of calu-1 are 120, 230, 380, 280, 240, respectively. Based on the data above, the Ct value differences between act-1 and calu-1 should be around 3.3, 2.5, 2.5, 1.9, 1.6,  respectively. Your numbers are very different from these data. Since the primers used are different, the values might change. But the change should be in a reliable range.

2. Fig.2 protein expression. Co-marker missing so it’s hard to claim the neurons you observed are AIA. The organism you observed is ER. You need dye the worm with AIA marker or ER-tracker, show the colocalization. Also, previous papers published that calu-1 expressed in AFD, GLR neurons (doi:10.1101/gr.271791.120;  doi:10.1016/j.jbc.2021.101094). We know by use different length of promoter might show different expression, please clarify the difference.

3. Dauer formation. No mechanism. This data showed calu-1  is required for dauer formation in C. elegans. We know insulin pathway genes: daf-16 and daf-2 are required for dauer formation. Is calu-1 regulates dauer formation through insulin pathway(calu-1 daf-16 double KO mutant)? Or just through the calcium metabolism? Does calcium supplementary plates rescues the dauer phenotype? No answer for any of these questions.

Minor points:

1. Calu-1 should be italic, for example line 74, 94, 104, 107, 144;

2. Line 120: paik should be park I think.

Round 2

Reviewer 1 Report

I accept revision of the manuscript from authors, however, the name of the strain ( of daf-16(mu86) (probably CF1038) I has not been included. Please check

Author Response

Thanks for the comment. We included the strain name of daf-16(mu86) I. 

Reviewer 2 Report

The author revised this manuscript very well

Author Response

Thanks for helpful review of our manuscript.

Reviewer 3 Report

I accept the comments from the authors about the first and second questions. However, the third question needs to be addressed more. Without double mutants such as daf-2;calc-1 or calcium supplementary experiments. I don't think this manuscript meets the Biology journal's standard for publication.

Author Response

à Calcium is an absolutely critical ion in a variety of signaling pathways including chemosenstaion by worm chemosensory neurons. It is therefore not surprising to imagine a role of calcium in signaling mediating dauer formation, as chemical sensing of environmental cues in chemosensory neurons is the first step in initiating the signaling. Indeed, as previously described, various chemosensory neurons that mediate dauer formation are related to calcium concentration (doi: 10.1895/wormbook.1.123.1; doi: 10.1126/science.1176331; doi: https://doi. org/10.1101/2021.08.17.456617). In particular, it was shown that calcium concentration increased in AIA interneuron when acute inhibition of AIA by pheromone occurred (doi: 10.1016/j.cub.2022.03.077). Since calumenin is a protein present in the ER, a calcium reservoir of cells and the site of post-translational modification of proteins, we believe that calumenin may be involved in multiple ways: by regulating calcium release and storage or by modulating key proteins mediating dauer switching or by regulating both. Therefore, elucidating the detailed molecular mechanisms could be another big story that doesn’t end with one or two figures. We believe that our results met the requirements of a “Brief reports”, an article type defined as a short, observational studies reporting preliminary results or a short complete study or protocol. We hope report soon the molecular mechanisms of how calu-1 functions for normal entry to diapause in C. elegans.

Round 3

Reviewer 3 Report

Brief reports”